# Mucin (MUC) Family Influence on Acute Lymphoblastic Leukemia in Cancer and Non-Cancer Native American Populations from the Brazilian Amazon

**DOI:** 10.3390/jpm12122053

**Published:** 2022-12-13

**Authors:** Angélica Leite de Alcântara, Lucas Favacho Pastana, Laura Patrícia Albarello Gellen, Giovana Miranda Vieira, Elizabeth Ayres Fragoso Dobbin, Thays Amâncio Silva, Esdras Edgar Batista Pereira, Juliana Carla Gomes Rodrigues, João Farias Guerreiro, Marianne Rodrigues Fernandes, Paulo Pimentel de Assumpção, Amanda de Nazaré Cohen-Paes, Sidney Emanuel Batista Dos Santos, Ney Pereira Carneiro dos Santos

**Affiliations:** 1Oncology Research Nucleus, Universidade Federal do Pará, Belém 66075-110, PA, Brazil; 2Human and Medical Genetics Laboratory, Instituto de Ciências Biológicas, Universidade Federal do Pará, Belém 66077-830, PA, Brazil

**Keywords:** MUC family, ALL, susceptibility, native American populations, Brazil

## Abstract

The mucin (MUC) family includes several genes aberrantly expressed in multiple carcinomas and mediates diverse pathways essentials for oncogenesis, in both solid and hematological malignancies. Acute Lymphoblastic Leukemia (ALL) can have its course influenced by genetic variants, and it seems more frequent in the Amerindian population, which has been understudied. Therefore, the present work aimed to investigate the MUC family exome in Amerindian individuals from the Brazilian Amazon, in a sample containing healthy Native Americans (NAMs) and indigenous subjects with ALL, comparing the frequency of polymorphisms between these two groups. The population was composed of 64 Amerindians from the Brazilian Amazon, from 12 different isolated tribes, five of whom were diagnosed with ALL. We analyzed 16 genes from the MUC family and found a total of 1858 variants. We compared the frequency of each variant in the ALL vs. NAM group, which led to 77 variants with a significant difference and, among these, we excluded those with a low impact, resulting in 63 variants, which were distributed in nine genes, concentrated especially in MUC 19 (*n* = 30) and MUC 3A (*n* = 18). Finally, 11 new variants were found in the NAM population. This is the first work with a sample of native Americans with cancer, a population which is susceptible to ALL, but remains understudied. The MUC family seems to have an influence on the development of ALL in the Amerindian population and especially MUC19 and MUC3A are shown as possible hotspots. In addition, the 11 new variants found point to the need to have their clinical impact analyzed.

## 1. Introduction

The mucin (MUC) family is a group of highly glycosylated macromolecules, abundantly expressed in mammalian epithelial tissue, whose primary functions include protecting and lubricating epithelial surfaces, providing the mucus gel-like structural properties, and contributing to intra- and intercellular signaling, cell proliferation, growth, and apoptosis [1,2]. They are encoded by several genes, many of which are associated mainly with solid neoplasms, especially regarding the gastrointestinal tract, as in colorectal cancer, gastric cancer, and pancreatic cancer. They serve as tumor markers, as in MUC1 for epithelial lineage cancers and MUC 16, which encodes the CA125 antigen, widely used in monitoring patients with ovarian cancer [3,4,5,6,7,8,9,10].

Nevertheless, the MUC family also seems to be aberrantly expressed in hematological neoplasms [11,12] and MUC 1 appear to have an essential role in leukemia stem-cell function, the induction of reactive oxygen species, and the promotion of terminal myeloid differentiation, being an attractive therapeutic target in hematologic malignancies [13,14].

Among the hematological neoplasms, we can mention Acute Lymphoid Leukemia (ALL), which comprises a set of lymphoid neoplasms morphologically and immunophenotypically similar to precursor cells of the B and T lineages [15,16]. It is the most common malignant neoplasm in childhood, with the highest risk being in children younger than five years old and accounting for 25–30% of all pediatric cancers worldwide [17,18,19].

According to data from the Brazilian National Cancer Institute (INCA), estimates indicate that new cases of leukemia expected for Brazil, for each year of the triennium 2020–2022, will be 5920 in men and 4890 in women. Except for non-melanoma skin cancers, leukemias rank fifth in number of cases in the northern region of Brazil (4.45/100,000). The northeast region ranks seventh (5.02/100,000), followed by the southeast (5.70/100,000) and midwest (4.29/100,000) regions [20].

The course of ALL can be influenced by several polymorphic variations [21,22,23] and studies point that ethnically diverse groups have fluctuations in the allele frequencies of related genomic variants, so that ethnic differences are able to influence the course of this disease [24,25]. The Brazilian population is one of the most heterogeneous, as a consequence of five centuries of miscegenation among three ancestral geographical groups: Amerindians, Europeans, and Africans, which show great genetic diversity within themselves; this could imply high fluctuations in the frequencies of important polymorphisms [23,26].

Different studies have shown that populations of Amerindian origin and mixed race with them have a higher risk of developing ALL, as well as a worse prognosis of this neoplasm [21,24,27]. In general, all Brazilian regions have a significant Amerindian genomic contribution, but the northern region has the highest one, with around 30% of the population being of the average Amerindian ancestry [28]. Besides Brazil, other American countries have high indigenous ancestry [29,30,31], however, there are currently few studies investigating the role of these important genes in worldwide Amerindian populations.

Thus, we aimed to investigate the MUC family exome in Amerindian individuals, in a sample containing healthy Amerindians and indigenous with ALL, in order to search for potential new or more frequent variants in this population and to investigate possible differences between molecular profiles of healthy indigenous subjects and those with cancer.

## 2. Materials and Methods

### 2.1. Study Population and Ethics

The population in our study consists of 64 Amerindians from Brazilian Amazon, which represent 12 different ethnic groups. Among them, five are children exclusively with B-cell ALL, who will compose the ALL group. These Amerindians with ALL were diagnosed and treated in the Ophir Loyola Hospital and the Octavio Lobo Childhood Oncology Hospital, two public hospitals specialized in childhood cancers, both located in the city of Belém, in Pará state, Northern Brazil.

The National Research Ethics Committee (CONEP) has approved this study, identified by No 1062/2006 and 4.568.181/2021. If needed, a translator explained the project and then all tribe leaders, who were all literate, signed a free-informed consent. The CONEP states that communities whose group culture recognizes the authority of the leader or the collective over the individual, obtaining authorization for research must respect this particularity, without prejudice to individual consent [32].

Their materials were collected according to the Declaration of Helsinki.

### 2.2. Selection of Genes

We analyzed 16 genes from the MUC family (MUC1, MUC2, MUC3A, MUC4, MUC5AC, MUC5B, MUC6, MUC7, MUC12, MUC13, MUC15, MUC16, MUC17, MUC19, MUC20, and MUC21). A total of 1858 variants were found, on which the following selection criteria were applied: (a) the read should be high coverage, with a minimum of 10 reads coverage (fastx_tools v.0.13-http://hannonlab.cshl.edu/fastx_toolkit/), consulted in 1 July 2021; (b) the predicted impact should be “modifier”, “moderate”, or “high” according to SNPeff (https://pcingola.github.io/SnpEff/, accessed on 1 July 2021); (c) the difference in allelic frequency of the variants between healthy indigenous individuals and indigenous children with ALL should be significant (*p*-value ≤ 0.05). The subsequent analyses were targeted to 63 variants that met all applied selection criteria.

### 2.3. Extraction of the DNA and Preparation of the Exomes

Samples of 5 mL of peripheral blood were collected from each of the participants of the study. The genetic material was extracted from these blood samples using the Roche Applied Science DNA extraction kit (Roche, Penzberg, Germany), following the manufacturer’s instructions. The technologic principle used was nucleic acid capture by glass fiber fleece immobilized in a special plastic filter tube and subjected to centrifugation [33]. In a solid-phase extraction, the magnetic bead method is used, so that DNA can be isolated easily from specimens by removing proteins and cellular debris on the beads [34,35]. It was quantified using a NanoDrop 1000 spectrophotometer (NanoDrop Technologies, Wilmington, DE, USA).

The exome libraries were created using the commercial Nextera Rapid Capture Exome kit (Illumina^®^, San Diego, CA, USA) and the SureSelect Human All Exon V6 kit (Agilent^®^, Santa Clara, CA, USA), also following the manufacturer’s protocol. The sequencing reactions were run in the NextSeq 500^®^ platform (Illumina^®^, San Diego, CA, USA) using the NextSeq 500 high-output v2300 cycle kit (Illumina^®^, San Diego, CA, USA).

### 2.4. Bioinformatics and Statistical Analysis

The bioinformatic analyses followed the approach described by Ribeiro-Dos-Santos et al. [36] and Rodrigues et al. [37]. Thus, low-quality reads were eliminated from the sequences and then the reference genome (GRCh38) was used to map and align using BWA v.0.7. The alignment was then processed to recalibrate the mapping quality, remove duplicate sequences, and finalize the local realignment. The results were processed in GATK v.3.2 in order to identify the reference genome variants. The annotations of the variants were analyzed by The Viewer of Variants (ViVa^®^Providence, USA) software. The variants were annotated in three databases—SnpEff v.4.3.T, Ensembl Variant Effect Predictor (Ensembl version 99), and ClinVar (v.2018-10)—and the in silico prediction of pathogenicity used the following databases: the SIFT (v.6.2.1), PolyPhen-2 (v.2.2), LRT (November, 2009), Mutation Assessor (v.3.0), Mutation Taster (v. 2.0), FATHMM (v.2.3), PROVEAN (v.1.1.3), MetaSVM (v1.0), M-CAP (v1.4), and FATHMM-MKL.

For statistical purposes, the cancer-free Amerindians were assigned to the native Amerindians (NAM) group, while the ALL patients were in the ALL group. The R v.3.5.1 program ran all the analyses. The Fisher’s exact test was used to evaluate the differences in the allelic frequencies between the NAM and ALL groups. A *p*-value ≤ 0.05 significance level was considered for all the analyses.

## 3. Results

At the beginning of the study, we had 1858 variants for the 16 genes of the analyzed MUC family. After filtering for sample quality, only 743 remained, which are displayed in Appendix A. Among them, we compared the frequency of each one in the ALL vs. NAM groups, in which we obtained 77 with a statistical difference. Finally, among these, we excluded those with a low impact, resulting in 63, which are arranged in Table 1.

The 63 polymorphisms were distributed in nine genes; 5 in MUC 16; 2 in MUC 17; 30 in MUC19; 1 in MUC2; 1 in MUC21; 18 in MUC 3A; 3 in MUC 4; 1 in MUC 5A, and 2 in MUC5B. This distribution can be visualized graphically in Figure 1.

Only 4 of these mutations were more frequent in the NAM group compared to the ALL group (rs60568788; rs2547065; rs10422567; and rs1559172) and 55 were unique to the ALL group, among them 36 being in coding sequence (CDS) region, 26 intronic, and 1 in “OTHER” (which refers to 5’UTR, 3’UTR, and intergenic regions).

In addition to the 77 variants with statistical differences between the NAM vs. ALL groups, within the group of 743 variants that passed the quality filter in the sample, 11 new variants were found in our healthy indigenous population, which are arranged in Table 2.

Among such 11 variants never described in the literature, they were found in 7 genes within the Amerindian population, all of them are present in heterozygosity in 12 different subjects, with two of the NAMs having two concomitant mutations. Two of them had high impact, two modifier, five moderate, and two low. The two high impact variants occurred in frame shift regions and were of the indel type.

## 4. Discussion

In our study, we found 743 variants with quality, among which 77 whose frequencies differed between healthy Amerindians and the ones with ALL, with 63 variants remaining after excluding those with a low impact. Most of them were concentrated in the MUC19 and MUC3A genes, suggesting them as potential hot spot regions in the genome, 55 mutations were unique to Amerindians with ALL—acting as possible biomarkers for ALL’s risk—and four variants were more frequent in the healthy Amerindian population-behaving as possible protective factors for the disease. In addition, 11 new variants were found in our healthy Native American population.

This is the first work to evaluate the exome of indigenous patients with ALL in comparison to healthy Amerindian individuals, an understudied population but apparently more susceptible to this pathology [24,38].

The MUC family is a group of highly glycosylated macromolecules found in mammalian epithelial tissue, consisting of several genes, among which many are associated mainly with solid neoplasms, especially the ones of the gastrointestinal tract [3,4,5,6,7], even though it also seems to be aberrantly expressed in hematological neoplasms [11,12], with MUC 1 being an attractive therapeutic target in hematologic malignancies [13,14]. Those findings are supported by our results, once they point to a possible influence of the MUC family on the course of ALL in Amerindian populations from the Brazilian Amazon.

Our study suggested MUC 19 and 3A as possible hot spots for ALL’s susceptibility, given that 76.5% of the variants with a significant difference were concentrated in these two genes. The GWAS studies conducted on the MUC 19 gene have already associated it with inflammatory bowel disease, Chron’s disease, and Parkinson’s disease [31,39,40,41], in addition to studies suggesting an association with clinical benefits in non-small cell lung cancer [42], carcinogenesis in breast cancer [43] and neuroblastoma [44]. MUC 3A is associated with disorders such as cap polyposis [45,46], colorectal cancer [47], and mucoepidermoid carcinoma of the lung [48]. Both genes appear to be related to the colorectal cancer pathway and their clinical implications are well described and relate to the overall function of the mucin family, however, there are no previous data linking these two genes to ALL.

When comparing the NAM vs. ALL groups, among variants with statistical significance we noticed that four of them (rs60568788; rs2547065; rs10422567; and rs1559172) were more frequent in the healthy population in comparison to the ones diagnosed with ALL, acting as possible protective factors. The variants rs60568788, rs10422567, and rs1559172 have no data described in the literature, while rs2547065 has been cited as a possible risk factor for epithelial ovarian cancer [49]. Similarly, 55 variants were unique to the group of Amerindians with ALL, suggesting them as possible risk factors, but only two of them have been cited in the literature, rs2857476 being associated with susceptibility to pulmonary fibrosis and rs13095016 with neurological complications following West Nile virus infection [50,51]. Their roles in the development of ALL need further clinical studies to be clarified.

We had two high impacts mutations whose frequencies differed between the NAM and ALL groups. The first one (rs5797672 in MUC 19) was found only in patients with ALL, even though they were a smaller sample composed by only five individuals, which shows that it could potentially impact the development of leukemias. The second high-impact mutation (rs60568788 also in MUC 19) was found in both the ALL-carrier and NAM groups, being more frequent in healthy indigenous subjects, which could be due do the fact that they were a larger sample. Both have not yet been cited in the literature, nor have had their clinical impact described.

Finally, we found 11 new variants in 7 different genes, 2 of them were indel-type frame shifters, with potential high clinical impact. One of them occurred in MUC 16, which corresponds to the CA125 antigen [8], whose features include a high proline, threonine, and serine, containing a possible transmembrane region and a potential tyrosine phosphorylation site. The CA125 is reported to have and increased expression in ovarian tumors over the past three decades, being the only clinically reliable diagnostic marker for ovarian cancer [52]. This gene has also been cited as (1) a poor prognostic marker for pancreatic, colon, and stomach cancers [6]; (2) involved in tumorigenesis and metastasis of lung cancer cells [53,54], and (3) contributing to the metastasis of pancreatic ductal adenocarcinoma [55].

The other high-impact new variant was found in MUC 17, whose expression is predominantly in the apical region of mature intestinal absorptive epithelial cells [56], being highly expressed in the adult small intestine, followed by the stomach and colon [57]. It has been shown to be highly expressed in gastric cancer, with a favorable prognosis for patients [58], with therapeutic potential. It has also been associated with ovarian and breast cancer [59,60,61]. However, neither MUC 16 nor MUC 17 has ever been associated with hematologic neoplasms.

Our hypothesis is that those novel variants are germline mutations, since they were described in a healthy population, so that leukemia would not justify them as being somatic mutations from diseased tissue. The occurrence of these mutations in the Amerindian population endorses the importance of studying them, since this is a genetically isolated population, and the clinical impact of such findings still needs to be investigated in further studies.

Thus, this is the first work in the literature to perform investigative exome studies in Amerindians patients, with a sample of both healthy and ALL-carriers indigenous from Brazilian Amazon, given that this population seems more susceptible to ALL but remains understudied. The MUC family may have a potential influence on the development of ALL in the Amazonian indigenous population, specially MUC19 and MUC3A, which are shown as possible hotspots. Furthermore, the 11 new variants found endorse the need to investigate their effects in human patients to assess their clinical impact.

Considering that Native American indigenous people live far from urban centers, there is a difficulty in accessing them and having them as part of investigations, however, given the importance of these findings in a context of data scarcity in isolated communities, we suggest that new studies should be conducted within those subjects, aiming to increase the statistical power by expanding the sample size and validating the results, ideally in a case–control study, which would best fit the purpose of clarifying the questions raised in this paper. Additionally, this type of study can assist public policies aimed at isolated native American people, as well as societies with high levels of admixture with those indigenous groups, in order to individualize and to improve the health care provided for these populations.

## Figures and Tables

**Figure 1 jpm-12-02053-f001:**
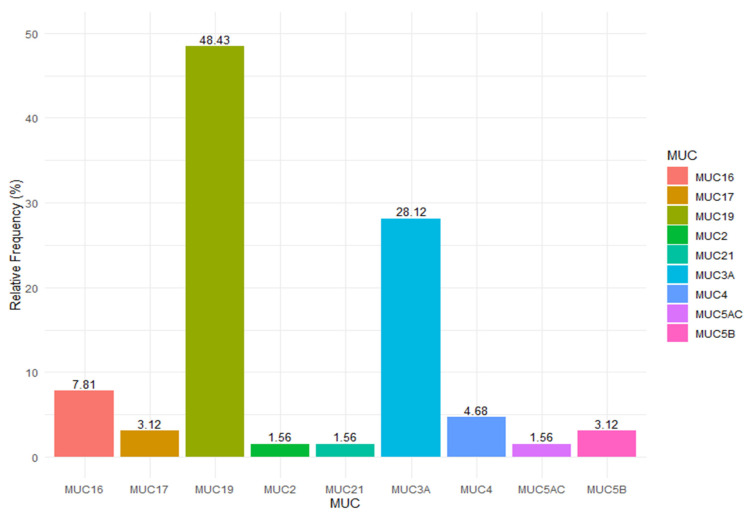
Frequency distribution of significant polymorphisms in each gene.

**Table 1 jpm-12-02053-t001:** Variants with significant difference between ALL vs. NAM groups.

MUC	Rs	Region Detailed	Var Type	Impact	ALL	NAM	*p*-Value
**MUC19**	rs5797672	FRAME_SHIFT	INDEL	HIGH	0.6	0	<0.01
**MUC19**	rs60568788	FRAME_SHIFT	INDEL	HIGH	0.6	0.71	<0.01
**MUC19**	rs1444222	INTRON	SNV	MODIFIER	0.6	0	<0.01
**MUC19**	rs4768284	INTRON	SNV	MODIFIER	0.6	0	<0.01
**MUC19**	rs994798	INTRON	SNV	MODIFIER	0.6	0	<0.01
**MUC19**	rs6581732	INTRON	SNV	MODIFIER	0.8	0	<0.01
**MUC19**	rs111256342	INTRON	INDEL	MODIFIER	0.8	0	<0.01
**MUC19**	rs1444215	INTRON	SNV	MODIFIER	0.9	0	<0.01
**MUC19**	rs2029615	NON_SYN	SNV	MODERATE	0.5	0	<0.01
**MUC19**	rs2638879	NON_SYN	SNV	MODERATE	0.5	0	<0.01
**MUC19**	rs2638875	NON_SYN	SNV	MODERATE	0.6	0	<0.01
**MUC19**	rs2638865	NON_SYN	SNV	MODERATE	0.6	0	<0.01
**MUC19**	rs2588401	NON_SYN	SNV	MODERATE	0.62	0	<0.01
**MUC19**	rs1492322	NON_SYN	SNV	MODERATE	0.6	0	<0.01
**MUC19**	rs1492313	NON_SYN	SNV	MODERATE	0.6	0	<0.01
**MUC19**	rs2251431	NON_SYN	SNV	MODERATE	0.7	0	<0.01
**MUC19**	rs2638866	NON_SYN	SNV	MODERATE	0.7	0	<0.01
**MUC19**	rs2933354	NON_SYN	SNV	MODERATE	0.8	0	<0.01
**MUC19**	rs2638868	NON_SYN	SNV	MODERATE	0.8	0	<0.01
**MUC19**	rs2405077	NON_SYN	SNV	MODERATE	0.8	0	<0.01
**MUC19**	rs7300780	INTRON	SNV	MODIFIER	0.4	0	<0.01
**MUC19**	rs75899846	INTRON	SNV	MODIFIER	0.4	0	<0.01
**MUC19**	rs73269929	INTRON	SNV	MODIFIER	0.4	0	<0.01
**MUC19**	rs78409264	NON_SYN	SNV	MODERATE	0.4	0	<0.01
**MUC19**	rs78204462	NON_SYN	SNV	MODERATE	0.4	0	<0.01
**MUC19**	rs73269926	INTRON	SNV	MODIFIER	0.4	0	<0.01
**MUC19**	rs7133943	INTRON	SNV	MODIFIER	0.4	0	<0.01
**MUC19**	rs6581782	INTRON	SNV	MODIFIER	0.4	0	<0.01
**MUC19**	rs79829730	INTRON	SNV	MODIFIER	0.4	0	<0.01
**MUC19**	rs73112046	NON_SYN	SNV	MODERATE	0.3	0	<0.01
**MUC3A**	rs75254397	INTRON	SNV	MODIFIER	0.5	0	<0.01
**MUC3A**	rs75196671	INTRON	SNV	MODIFIER	0.5	0	<0.01
**MUC3A**	rs78724937	INTRON	SNV	MODIFIER	0.5	0	<0.01
**MUC3A**	rs78470577	INTRON	SNV	MODIFIER	0.5	0	<0.01
**MUC3A**	rs73398800	INTRON	SNV	MODIFIER	0.5	0	<0.01
**MUC3A**	rs75547895	INTRON	SNV	MODIFIER	0.5	0	<0.01
**MUC3A**	rs73714242	NON_SYN	SNV	MODERATE	0.5	0	<0.01
**MUC3A**	rs78538898	NON_SYN	SNV	MODERATE	0.5	0	<0.01
**MUC3A**	rs76249962	NON_SYN	SNV	MODERATE	0.5	0	<0.01
**MUC3A**	rs78684063	NON_SYN	SNV	MODERATE	0.5	0	<0.01
**MUC3A**	rs28515787	NON_SYN	SNV	MODERATE	0.5	0	<0.01
**MUC3A**	rs74460367	NON_SYN	SNV	MODERATE	0.5	0	<0.01
**MUC3A**	rs78826835	NON_SYN	SNV	MODERATE	0.5	0	<0.01
**MUC3A**	rs79233494	NON_SYN	SNV	MODERATE	0.5	0	<0.01
**MUC3A**	rs73163757	NON_SYN	SNV	MODERATE	0.5	0	<0.01
**MUC3A**	rs73398732	NON_SYN	SNV	MODERATE	0.4	0	<0.01
**MUC3A**	rs78584246	NON_SYN	SNV	MODERATE	0.5	0.046	<0.05
**MUC3A**	rs75517157	NON_SYN	SNV	MODERATE	0.5	0.064	<0.05
**MUC5B**	rs2857476	INTRON	SNV	MODIFIER	0.8	0	<0.01
**MUC5B**	rs77287508	NON_SYN	SNV	MODERATE	0.5	0.02	<0.01
**MUC17**	rs6966570	INTRON	SNV	MODIFIER	0.4	0	<0.01
**MUC17**	rs10246021	INTRON	SNV	MODIFIER	0.5	0	<0.01
**MUC2**	rs12416873	INTRON	SNV	MODIFIER	0.6	0	<0.01
**MUC4**	rs13095016	NON_SYN	SNV	MODERATE	0.3	0	<0.01
**MUC4**	rs2259419	INTRON	SNV	MODIFIER	0.3	0	<0.01
**MUC4**	rs729593	NON_SYN	SNV	MODERATE	0.5	0	<0.01
**MUC5AC**	rs1132434	NON_SYN	SNV	MODERATE	0.6	0	<0.01
**MUC21**	rs2517418	INTRON	SNV	MODIFIER	0.3	0	<0.01
**MUC16**	rs3764552	INTRON	SNV	MODIFIER	0.3	0	<0.01
**MUC16**	rs2547065	NON_SYN	SNV	MODERATE	0.3	0.85	<0.01
**MUC16**	rs2591592	NON_SYN	SNV	MODERATE	0.6	0.14	<0.05
**MUC16**	rs10422567	NEXT-PROT	SNV	MODERATE	0.7	0.97	<0.05
**MUC16**	rs1559172	NON_SYN	SNV	MODERATE	0.7	0.96	<0.05

**Table 2 jpm-12-02053-t002:** New variants found in healthy indigenous population.

MUC	Chro	Position	Region	Region Detailed	Var Type	Impact	Refer	Variant	Freq
**MUC17**	chr7	101036821	CDS	FRAME_SHIFT	INDEL	HIGH	CT	C	0.018
**MUC16**	chr19	8949879	CDS	FRAME_SHIFT	INDEL	HIGH	TTGGA	T	0.009
**MUC5B**	chr11	1258072	intronic	INTRON	SNV	MODIFIER	C	A	0.036
**MUC5B**	chr11	1260321	intronic	INTRON	SNV	MODIFIER	G	C	0.042
**MUC16**	chr19	8949763	CDS	NON_SYNONYMOUS	SNV	MODERATE	T	G	0.009
**MUC16**	chr19	8952957	CDS	NON_SYNONYMOUS	SNV	MODERATE	G	T	0.009
**MUC21**	chr6	30987157	CDS	CODON_CHANGE + CODON_INSERTION	INDEL	MODERATE	A	AGCA	0.009
**MUC5AC**	chr11	1168745	CDS	NON_SYNONYMOUS	SNV	MODERATE	G	C	0.009
**MUC16**	chr19	8956261	CDS	NON_SYNONYMOUS	SNV	MODERATE	G	C	0.019
**MUC19**	chr12	40429590	CDS	SYNONYMOUS_CODING	SNV	LOW	A	G	0.009
**MUC6**	chr11	1032052	OTHER	SPLICE_SITE + SYNONYMOUS	SNV	LOW	G	A	0.011

## Data Availability

The authors confirm that the data supporting the findings of this study are available within the article and its Appendix A. Raw data of the studied genes are available from the corresponding author, upon reasonable request.

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
