# Peer review of "Mucin (MUC) Family Influence on Acute Lymphoblastic Leukemia in Cancer and Non-Cancer Native American Populations from the Brazilian Amazon"

_jpm, 2022, doi:10.3390/jpm12122053_

Round 1

Reviewer 1 Report

The authors describe MUC family mutations in healthy controls and ALL patients from Amerindian ethnicity. Although the number of ALL patients in the cohort is too small to claim that the 55 mutations identified in Amerindian with ALL can be used as potential biomarkers for ALL. But, it's a good start to an ethno-epidemiological study, where authors can base future studies to correlate their test-cohort findings to a larger patient cohort in order to derive more meaningful biomarkers.

Author Response

We thank Reviewer 1 for the comment and we agree that ours is indeed a small sample, however, amerindians are a rare cohort and amerindians with cancer are an even rarer one, so we reinforce the need for studies with a larger sample, and we agree that our research is a useful start to an ethno-epidemiological study in order to derive more meaningful biomarkers.

Reviewer 2 Report

Dear Authors,

Thank you for your contribution. I read your article on the influence of mucin (MUC) family on the development of acute lymphoblastic leukemia (ALL) in the Amerindian population. I think the study of cancer predisposition is a relevant topic in the pediatric and adult patients with ALL and it is important to investigate factors that can influence it. Indeed, the aim of your article is quite interesting, even if the number of patients compared are small, in particular the group of 5 patients with ALL.

I found a lack of consistence in your analysis since the two groups compared seem not to have similar features (the age, for instance). Moreover, in my opinion, the population analyzed is too small to be representative of Amerindian population.

 My comments after reviewing are the following:

Title:

-        I would explicit the abbreviation MUC also in the title

-   I would highlight the aim of the study to compare molecular profiles of healthy indigenous and those with cancer

Abstract:

-   line 18 I would describe in few words the characteristics of the population

Introduction:

-     line 35 I would put a full stop. The sentence is too long, and it is difficult to follow.

     line 37 how it can be used as tumor marker?

-        line 48 I would specify the medium age

Materials and Methods:

-        It is not specified the period of time of the study

-        I would explain the criteria of inclusion and exclusion

-        Is the study retrospective o prospective?

-        line 96 ALLL – please erase one of the L

Results

-        line 136 please explain in which way you define ‘LOW impact’

-        line 147 CDS region. Please explain the abbreviation.

-     line 147 it is not clear the meaning of OTHER. I would define in the text or in the table.

    Discussion

-        line 169 pathology1. Type mistake, I guess.

-   line 230 seems more susceptible to ALL’ – please explain the reasons of this sentence

Kind regards

Author Response

We would first like to thank Reviewer 2 for the comments, which much helped to enhace our work. 

As for the title, we suggest "Mucin (MUC) family influence on Acute Lymphoblastic Leukemia in cancer and non-cancer Native American populations from the Brazilian Amazon" as an alternative to focus on the comparison between the molecular profiles of healthy indigenous and those with cancer, as suggested by the reviewer.   We have altered the abstract and the introduction to meet the reviewer's requests, and all our chances are highlighted in yellow in the attached archive.   As for the Methods section, our ethics committee has approved GWAS Study in ALL patients in 2021, however, our work within the indigenous populations has been on course since 2006 and it remains till these days. It is a retrospective study and, as amerindians with cancer are a rare population, we included all patients diagnosed with ALL at time of collection who consented to the participation in our study, with a total of five. The inclusion of healthy indigenous, considering that these are hard to reach populations, consisted in the ones who were healthy at the time of data collection. As an exclusion criterion, we excluded indigenous people with very close kinship.    As for the Results section, we point out lines 101-103 "the predicted impact should be "modifier", "moderate" or "high" according to SNPeff (https://pcingola.github.io/SnpEff/)". SNPeff uses in silico computational methods that assess the impact based on the position of the variant and the type of the variant that has been altered. "CDS" region is the "CoDing Sequence" region, this information has been added to the text. The "OTHER" region refers to 5'UTR, 3'UTR and intergenic regions, which has been also added to text.    Finally, regarding the discussion section we point that the amerindians increased susceptibility to ALL has been described in both introduction and discussion sections, as in Lines 66-67: Different studies have shown that populations of Amerindian origin and mixed race with them have a higher risk of developing ALL, as well as a worse prognosis of this neoplasm (references 21,24, and 27)  Besides that, Quiroz et al indicate that higher incidence rates and poor outcomes have been reported among Latin American patients (Latinos) with acute lymphoblastic leukemia (ALL).   Once more, we thank you for your time and reinforce that we are at your disposal for any further comments.   Best regards, Ney Santos.   

Round 2

Reviewer 2 Report

Dear Authors,

I read your new version of the article. Thank you for your effort to cover the lacks I pointed out previously.

In my opinion it is now possible to publish your manuscript.